# Multidisciplinary Management of an Atypical Gigantic Sciatic Nerve Schwannoma: Case Presentation and Systematic Review

**DOI:** 10.3390/neurosci6040095

**Published:** 2025-09-28

**Authors:** Octavian-Mihai Sirbu, Mihai-Stelian Moreanu, Mark-Edward Pogarasteanu, Andreea Plesa, Mihaela Iordache, Teofil Mures, Anca Maria Sirbu, Marius Moga, Marian Mitrica

**Affiliations:** 1Doctoral School, Faculty of Medicine, “Carol Davila” University of Medicine and Pharmacy, 050474 Bucharest, Romania; octavian-mihai.sirbu@drd.umfcd.ro (O.-M.S.); andreea.plesa@drd.umfcd.ro (A.P.); 2Clinical Neurosciences Department, “Carol Davila” University of Medicine and Pharmacy, 050474 Bucharest, Romania; marian.mitrica@umfcd.ro; 3Department of Neurosurgery, “Dr. Carol Davila” Central Military Emergency University Hospital, 010825 Bucharest, Romania; 4Faculty of Medicine, “Carol Davila” University of Medicine and Pharmacy, 8 Eroii Sanitari Boulevard, 050474 Bucharest, Romania; mark.pogarasteanu@umfcd.ro (M.-E.P.); marius.moga@umfcd.ro (M.M.); 5Department of Orthopaedics and Trauma Surgery, “Dr. Carol Davila” Central Military Emergency University Hospital, 010242 Bucharest, Romania; 6Department of Neurology, “Dr. Carol Davila” Central Military Emergency University Hospital, 010825 Bucharest, Romania; 7Medicala I Clinic, “Dr. Carol Davila” Central Military Emergency University Hospital, 010242 Bucharest, Romania; mihaela.iordache@scumc.com; 8Faculté de Santé, Sorbonne Université, Les Cordeliers, 15 Rue de l’École de Médecine, 75006 Paris, France; tcmures@for.paris; 9National Institute of Medical Expertise and Recovery of Working Capacity, Street Panduri 22, 050659 Bucharest, Romania; anca-maria.sirbu@drd.umfcd.ro

**Keywords:** giant sciatic nerve schwannoma, intracapsular resection, intraoperative nerve monitorization, biopsy

## Abstract

Background: Sciatic nerve schwannomas are rare benign tumors that can develop along the nerve’s course, from the pelvis to the thigh. Giant schwannomas, defined as those exceeding 5 cm, are particularly rare and may alter the tumor’s anatomical relationship with the nerve, impacting surgical strategy. Methods: A PRISMA 2020-compliant systematic review was conducted using the terms (“sciatic” AND “schwannoma”) for publications from 2000 to October 2024. Of 166 identified articles, we excluded those lacking giant schwannoma cases or involving syndromic associations. We also report a novel case from our center. Results: Our patient, a 35-year-old woman, presented with tingling and discomfort while sitting, localized to the left thigh, without radicular pain or motor deficits. MRI revealed a 14 × 7 cm mass. This is, to our knowledge, the first reported case of a giant solitary sciatic schwannoma of these dimensions located exclusively in the thigh, resected via intracapsular dissection with nerve monitoring, that was fully documented and reported. The review yielded 22 relevant articles, most involving pelvic or pelvic–thigh junction locations, with low recurrence rates. Conclusions: Giant sciatic schwannomas may be asymptomatic and slow-growing. This case is notable for tumor’s location, large size, and successful nerve-sparing surgical outcome.

## 1. Introduction

Schwannomas are benign, slow-growing, encapsulated tumors originating from Schwann cells of neuroectodermal origin. They may develop along the course of any peripheral or cranial nerve (III–XII) with myelin produced by Schwann cells. Peripheral schwannomas represent the most common type of nerve sheath tumor, with an annual incidence of 3–4 cases per million, accounting for 5–8% of all soft tissue tumors. They show no sex predilection and typically present between the ages of 20 and 50 years [1,2]. Recurrence following resection is rare, except in syndromic cases such as neurofibromatosis type 2 (NF-2), schwannomatosis, or multiple endocrine neoplasia [3]. Peripheral schwannomas occur more frequently in the upper extremities compared to the lower extremities, with sciatic nerve involvement being particularly rare. The risk of malignant transformation is estimated at approximately 1% [1,4]. Morphologically, schwannomas may present as solitary encapsulated tumors, or more rarely as plexiform or clustered lesions. Clinical presentation is variable, ranging from asymptomatic cases to those with radiating pain, numbness, or motor deficits. Although tumor size is often cited as a determinant of clinical symptoms, several reports, including the present case, describe giant schwannomas without preceding symptoms [1]. It is likely that anatomical location and the degree of neural compression are more relevant to clinical presentation, and schwannomas should be considered in the differential diagnosis of discogenic and non-discogenic limb pain. Microsurgical enucleation under direct intraoperative nerve stimulation, with meticulous dissection of nerve bundles, remains the treatment of choice.

We present the case of a solitary giant sciatic nerve schwannoma (>5 cm) that was completely resected, with no recurrence at the 6-month follow-up. This case is notable due to several unique features: the tumor’s extraordinary size (15 cm) despite confinement to the thigh, its uniformly firm and homogeneous consistency without cystic degeneration, and the absence of clinical symptoms despite its dimensions. The thigh is rarely permissive of such silent volumetric expansion, underscoring the rarity of this presentation. Comparable reports are limited to a single case in the literature, where a similar lesion occurred in an elderly patient, possibly reflecting a protracted course of tumor development [4].

A further point of interest lies in the patient’s known Turner syndrome (TS), a chromosomal disorder characterized by gonadal dysgenesis and chronic estrogen and progesterone deficiency. These hormones are known to influence Schwann cell biology, particularly in regulating proliferation, differentiation, and myelin synthesis through receptor-mediated and paracrine mechanisms. Previous studies have shown that elevated levels of estrogen and progesterone can stimulate Schwann cell activity and have been implicated in the pathogenesis of neurogenic tumors, including neurofibromas and vestibular schwannomas. Conversely, the chronic hypogonadal state in TS disrupts these conventional pathways, theoretically reducing the risk of hormone-dependent tumorigenesis. The occurrence of a giant schwannoma in such a hormonal context therefore represents a paradox and has not, to the best of our knowledge, been previously documented. This unique clinical intersection not only adds to the significance of the present case but also raises important questions about alternative tumorigenic mechanisms—such as inflammatory signaling, angiogenic pathways, or compensatory growth factor activity—that may be unmasked in the absence of sex steroid regulation.

In addition to this illustrative report, we conducted a systematic review of the literature on solitary giant sciatic nerve schwannomas. The review consolidates published evidence, highlights clinical and radiological patterns, and evaluates management strategies and outcomes. By synthesizing current knowledge, this work provides the most comprehensive overview to date of these rare tumors, offering practical guidance for clinicians confronted with diagnostic or therapeutic dilemmas in this context.

## 2. Illustrative Case

A 35-years old woman was admitted to the hospital with a history of 8 months’ distension of the left thigh, discomfort and paresthesia only when sitting. From his past medical background, we mentioned that the patient was known for Turner syndrome and took on a regular basis anti-conception medication. At neurological examination, the patient presented with no signs of radiating pain, no motor deficits or other neurologic symptoms. At close palpation, there was a solid mass on her left posterior thigh, around 15 cm, tender to palpation, but no signs of swelling or color change were present.

### 2.1. Histological Examination

Considering this suspicious evolution, our first attempt was to perform a stereotactic biopsy of the lesion, and histology examination revealed sciatic nerve schwannoma. After the biopsy no symptom increase appeared. On microscopic examination, there were identified spindle cells arranged in intersecting bundles, with palisading areas, vessels with thickened walls, presence of foamy macrophages, and focally enlarged hyperchromic nuclei, with Anthony A and B type areas.

### 2.2. MRI Aspect

MRI examination revealed a solid, oval-shaped formation measuring approximately 14 cm in its long diameter and 7 cm in its axial diameter, appearing strongly hyperintense on T2WI (T2-weighted images), and showing the same intensity on T1WI with muscles (Figure 1). The tumor extended in the 1/3 middle and 1/3 distal compartment of the left thigh, with compressive effect to the adjoining tissues, being localized between *Adductor Magnus muscle*, anteriorly, *Semimembranosus* and *Semitendinosus muscle* on the medial site, and *Biceps femoris* (*caput longus*) on the posterior and lateral site.

### 2.3. Operative Course

After a thorough discussion of the risks and benefits, the patient consented to surgical resection of the tumor. The procedure was performed with the patient in the prone position under continuous intraoperative neurophysiological monitoring (INM). Needle electrodes were placed in the gastrocnemius, peroneus, anterior tibialis, and extensor hallucis longus muscles. We used the NIM Vital™ Nerve Monitoring System (Medtronic, Minneapolis, MN, USA). Muscle responses to electrical stimulation were evident throughout the procedure, producing contractions strong enough to cause the foot to strike the table’s metal support. Prior to reaching the capsule, stimulation was performed at 2 mA; during intracapsular dissection, the amplitude was reduced to 0.5 mA to allow precise identification and dissection of individual nerve fibers.

Surgery was initiated through a posterior longitudinal incision in the middle-to-distal third of the thigh. Careful stepwise dissection revealed the giant tumor capsule (Figure 2A). No muscle was incised or retracted, as the tumor had expanded toward the surface, displacing the surrounding tissue. The biceps femoris and semitendinosus muscles were compressed by the mass. After capsule identification, a nerve stimulator was applied to locate a non-responsive area for incision. The capsule appeared reddish, highly vascularized, and adherent to the underlying tumor, which had a solid consistency. Both the proximal and distal poles of the tumor were identified and carefully dissected (Figure 2B). Determining the relationship between the tumor and the parent nerve was essential for planning safe removal. In this case, the tumor was suspended from the nerve in a hammock-like fashion, with only a few fibers incorporated into the capsule. Stimulation of these fibers elicited gastrocnemius contraction with plantarflexion, confirming their functional integrity. The presumed origin fiber, completely enveloped within the tumor, was divided (Figure 2C). The tumor was then removed en bloc, measuring 14.5 × 7 × 6.4 cm, and submitted for histopathological analysis (Figure 2D).

Postoperative evolution was unremarkable, without any perioperative complications. Yet, it should be mentioned that the patient developed S1 motor root weakness at walking on tiptoes. The paresthesia disappeared after the operation, and the patient was discharged at day 3. No recurrence was observed at the 6-month follow-up, and the deficit had improved. The patient currently attends regularly physical recovery.

### 2.4. Consent

Patient consent was obtained for inclusion in this case report. No identifying details have been disclosed. Data was obtained from medical history papers, morphology examinations, and MRI scanning.

## 3. Materials and Methods

A systematic review was conducted using PRISMA 2020 flow-diagram as showed in Figure 3. We searched in PubMed database using formula (“sciatic” AND “schwannoma”) from 2000 to October 2024 and found 166 articles matching the keywords. No duplicates were found. After initial screening, 58 articles were excluded based on their titles and abstracts that were not presenting case reports or series of cases and were about guidelines, morphology, imaging, or basic science. In the next step, among the reports sought for retrieval, 29 articles were excluded due to the absence of abstract or because they were not in English. Finally, exclusion criteria were applied. There were 16 articles reporting on other tumors or tumor-related syndromes, 9 cases of a schwannoma reported in other locations, 14 cases of a schwannoma under 5 cm, 14 cases of a schwannoma whose tumor dimension could not be identified and which were syndrome-related, and 4 cases of articles in which dimension was not mentioned. After reviewing all available data, 22 articles reporting a solitary giant sciatic nerve schwannoma were selected. Important data such as age, sex, clinical presentation, imaging, mass localization, postoperative course and follow-up period were compiled in Table A1, Table A2 and Table A3 using Excel 2016 functions (Appendix A) [1,2,4,5,6,7,8,9,10,11,12,13,14,15,16,17,18,19,20,21,22,23]. In cases where studies reported multiple measures (e.g., preoperative and postoperative symptoms at different time points), the most complete and clinically relevant results were collected, prioritizing final postoperative outcomes at the longest follow-up.

Given the rarity of solitary giant sciatic schwannomas, and the heterogeneous nature of the available reports (primarily single case reports and small case series), a narrative synthesis was chosen rather than a quantitative meta-analysis. This approach allowed for structured comparison of clinical presentation, imaging features, surgical approaches, and outcomes across individual cases, while acknowledging variability in reporting standards, follow-up duration, and patient characteristics. Studies where only ranges or descriptive terms were provided (e.g., “large” or “giant”) were excluded if precise measurements could not be identified, in accordance with predefined eligibility criteria. Similarly, when data regarding tumor location, presence of schwannomatosis, or association with neurofibromatosis were incomplete, such studies were excluded from the final synthesis to preserve consistency. Because the review focused exclusively on solitary giant sciatic schwannomas (>5 cm), only reports that included exact tumor dimensions were retained. The main factors contributing to variability among the included articles were differences in reporting detail, heterogeneity in tumor size, and variability in anatomical location along the sciatic nerve. Additional sources of heterogeneity included the duration of preoperative symptoms and differences in follow-up duration.

## 4. Discussions

### 4.1. Demographics

Since the first description of this tumor by Verocay in 1908, schwannomas have been referred to by various terms, including neurilemoma, neurinoma, perineurial fibroblastoma, and peripheral glioma. The lower limb is an uncommon site for a schwannoma, and sciatic nerve involvement is particularly rare, accounting for only ~1% of all reported cases [2,11]. The overall incidence of schwannomas is estimated at 3–4 cases per million individuals annually [2,11]. While previous reviews [19] have suggested that sciatic nerve schwannomas typically present between 20 and 50 years of age, our analysis demonstrates a broader distribution (21–79 years), with a mean age of 47.5 years. We also observed a female predominance (14:8, 63%), consistent with our present case.

Sciatic nerve schwannoma remains a rare entity. Yan et al. reported in 2022 that fewer than 40 cases had been described in the literature [7]. Although schwannomas may arise from any peripheral nerve of the lower limb, the posterior tibial nerve is most frequently reported [19]. Sciatic schwannomas can be classified as intrapelvic, extrapelvic, or combined intra–extrapelvic, the latter being more often associated with giant tumors [15]. In our review, 11 of 22 giant schwannomas (50%) were located in the abdominopelvic region, either purely pelvic or extending across both compartments. This distribution likely reflects two factors: (1) the pelvis provides more space for tumor expansion than the thigh, allowing tumors to reach giant dimensions without early detection, and (2) pelvic tumors often remain asymptomatic and are less accessible to palpation. Advances in ultrasound and MRI have, however, progressively shortened diagnostic delays [7]. Within the thigh, tumor distribution was approximately equal across proximal, middle, and distal levels (22–27%).

### 4.2. Clinical Presentation

Clinical manifestations depend primarily on the tumor’s relationship to adjacent nerve bundles and its location along the sciatic course. In our review, the most common symptom was radiating pain (12/22 cases, 54.5%), followed by discomfort or pain with palpation or sitting (9/22, 40.9%), and sensory disturbances such as paresthesia or hypoesthesia (7/22, 31.8%). Four patients (18.2%) were asymptomatic, three of whom had pelvic tumors. Motor weakness was uncommon, reported in only two cases (9%). Abdominal pain and defecation difficulty were noted in two patients with pelvic extension.

Palpation was feasible in 16/22 cases (72.7%), though only 9 of these were painful. Importantly, tumor size did not consistently correlate with symptom severity. Schwannomas characteristically displace rather than infiltrate nerve fibers, allowing late symptom onset. Indeed, our patient harbored a 15 cm thigh mass occupying two-thirds of the compartment without radiating pain, paresthesia, or motor deficits.

When diagnosing sciatica, clinicians must carefully consider non-discogenic causes. Although lumbar disc herniation is the most common etiology, differential diagnoses include sciatic neuritis, sacroiliitis, piriformis syndrome, hamstring tendinopathy, soft tissue tumors, and peripheral nerve sheath tumors. These conditions may coexist, complicating evaluation. Refractory sciatica—characterized by unremarkable lumbar MRI, absent straight leg raise sign, but positive Tinel’s sign—should raise suspicion of peripheral nerve pathology such as schwannoma [11]. Interestingly, schwannoma-related pain may also mimic distal entrapment syndromes (e.g., tarsal tunnel), as proximal compression of grouped sensory fibers can generate distal symptoms [23]. Intra-abdominal tumors may additionally compress lumbosacral roots, causing buttock, pelvic, or perineal pain, while lateral or posterior expansion may lead to sacroiliac pain, rectal compression, or constipation [20].

### 4.3. Tumor Morphology and Behavior

Schwannomas typically measure 1–3 cm; those exceeding 5 cm are classified as giant. Once termed “ancient schwannomas,” such tumors continue to be reported. Our review identified 22 reports on giant schwannomas published since 2000 [3], with schwannoma sizes ranging from 5 to 30 cm (mean long-axis diameter 9.3 cm). Our case measured 15 cm.

The rarity and clinical importance of giant schwannomas lies in their indolent growth, often silent progression, and morphological changes that complicate diagnosis. While classic schwannomas appear as solid, firm, encapsulated lesions, giant variants frequently exhibit degenerative changes, including cyst formation and necrosis. Consequently, their MRI signal becomes heterogeneous, broadening the differential diagnosis to include both benign and malignant tumors. Proposed mechanisms of degeneration include vascular insufficiency secondary to progressive enlargement, leading to necrosis, hemorrhage, and hyaline degeneration in Antoni B areas, with cysts sometimes predisposing to infection [12].

Grossly, sciatic schwannomas are solitary, encapsulated, eccentric to the nerve, and well circumscribed by perineurium and epineurial collagen. Cystic degeneration is common; intratumoral hemorrhage is rare. Plexiform schwannomas are exceedingly uncommon [24]. Histologically, they are composed of spindle-shaped Schwann cells arranged in a biphasic pattern: hypercellular Antoni A areas with nuclear palisading and Verocay bodies, and hypocellular Antoni B areas with loose stroma and lipid-laden cells. Immunohistochemistry is rarely needed [3]. Schwannomas are benign; malignant transformation is exceptional [4]. Unlike neurofibromas in NF1, which give rise to malignant peripheral nerve sheath tumors (MPNST) in ~10% of cases, there is little evidence linking schwannomas to malignant progression [8,25]. Rarely, malignant transformation has been reported, producing epithelioid MPNSTs or angiosarcomas [21]. Only one malignant giant schwannoma was identified in our review, suggesting size alone is not a determinant of malignancy, contrary to the prevailing notion in the literature that larger schwannomas may carry a higher risk of malignant transformation [26]. Instead, chronic vascular stasis and edema may contribute to sarcomatous transformation [21]. Malignant schwannomas are aggressive, with early pain and neurological deficits, requiring multimodal treatment and carrying a poor prognosis, in contrast to benign schwannomas, which are cured with surgical excision alone.

### 4.4. Imaging and Biopsy

Recent registry-based studies underscore the diagnostic complexity of rare peripheral nerve tumors and the essential role of multidisciplinary collaboration—including neuropathologists and radiologists—to differentiate such lesions from mimickers. Ultrasound is recommended as the first-line tool for superficial soft-tissue lesions due to its affordability, absence of radiation, high diagnostic accuracy, and ability to guide safe biopsies [27]. Sonographically, schwannomas appear as well-defined, hypoechoic, fusiform masses, typically eccentric to the parent nerve, whereas neurofibromas are centrally located [28]. In large, heterogeneous tumors, ultrasound-guided biopsy may be considered to exclude malignancy or infection prior to surgery [1,4,12,14]. However, biopsies are sometimes inconclusive, yielding only cystic fluid or septations [16]. In suspected infection, surgical resection is preferred over observation [12].

MRI remains the gold standard for sciatic schwannoma diagnosis, though ultrasound is useful. Schwannomas slowly displace the surrounding tissue, creating their own anatomical plane. On MRI, they usually appear isointense to muscle on T1-weighted images and hyperintense on T2-weighted sequences. Cystic or hemorrhagic degeneration produces heterogeneous signals. In our review, 9/22 cases (40.9%) exhibited such heterogeneity. The “target sign” (peripheral hyperintensity with central hypointensity) may be seen, but it is not specific, as it occurs in neurofibromas as well [1,12,15]. The “split fat sign,” reflecting preserved perineural fat, favors schwannoma over neurofibroma [28,29]. Another supportive feature is the “fascicular sign,” showing multiple small hyperintense rings on T2WI, corresponding to fascicles, more commonly observed in schwannomas [29]. MR neurography provides additional specificity in visualizing fascicular disruption or displacement [10]. The Neuropathy Score Reporting and Data System (NS-RADS) offers a standardized MRI-based framework for the evaluation of peripheral nerve sheath tumors. Its classification into N1–N4 categories provides a reproducible tool to differentiate benign from malignant or recurrent lesions [27].

Although rarely employed, for establishing a more accurate differential diagnosis and minimizing the risk of overlooking malignant transformation, high FDG uptake on FDG PET/CT can serve as a valuable tool for early detection of malignancy [26].

### 4.5. Treatment, Operative Course, Intraoperative Neuromonitoring, Postoperative Evolution

Classically, the surgical approach to sciatic schwannomas depends on tumor location: a longitudinal posterior incision for thigh lesions, or a transabdominal approach for pelvic schwannomas. In cases of giant schwannomas with both intra- and extrapelvic extension, a staged surgical strategy may be warranted [22]. The gold standard treatment remains gross total resection using microsurgical techniques under continuous intraoperative neurophysiological monitoring (INM) [4]. The initial step involves exposing the tumor through the least invasive approach possible, with particular attention to minimizing approach-related pain and morbidity. Once exposed, the tumor-bearing nerve should be identified proximally and distally, and, when feasible, looped circumferentially with rubber bands. After the removal of surrounding connective tissue or fat, the tumor surface is inspected to trace the course of functionally important fascicles, a step that requires magnification with a microscope or surgical loupes [30].

Surface stimulation is applied at multiple tumor sites, and motor or sensory responses are recorded to guide dissection. Multidisciplinary teams are optimally equipped to address peripheral nerve tumors.

A key step prior to resection is determining the tumor’s relationship to the parent nerve. Li et al. proposed a classification system: type I, where the nerve stem is displaced to one side, forming a “hammock” around the tumor, and type II, where the tumor is completely surrounded by nerve bundles [31]. Type I tumors can generally be dissected off the “hammock,” while type II tumors require delicate intracapsular dissection within epineurial layers. The decision between extracapsular versus intracapsular resection must, therefore, be made intraoperatively. While en bloc extracapsular resection is traditionally preferred for its efficiency, minimal bleeding, and radicality [22], it may not be feasible in type II tumors embedded within the nerve. In such cases, piecemeal resection with stepwise decompression is often safer. Capsular entry is performed at the site farthest from the functional fascicles. To facilitate this step, the tumor can often be gently rotated along its longitudinal axis. Following capsular incision, the dissection plane must be carefully identified; however, in giant schwannoma prior step of debulking is necessary.

In our case, the tumor was attached in a type I hammock-like configuration, enabling en bloc removal. Nonetheless, even in giant schwannomas, intracapsular resection carries the risk of neurological deficits, as fascicles may traverse the tumor. Intraoperative monitoring is indispensable in these circumstances, enabling the identification of functional fibers and distinction between motor and sensory components. Chronic compression may thin the nerve and impair function, making this assessment critical. This was demonstrated in a study using motor-evoked potentials MEPs, when intraoperative traction or compression of nerve fascicle resulted in lower signals of MEPs despite nerve trunk preservation, likely due to intraoperative ischemia. These findings highlight the importance of preserving perineural vessels and performing frequent INM [32].

Regarding INM, various modalities are available. Spontaneous electromyography (sEMG) records continuous myogenic activity from muscles innervated by nerves at risk. Burst discharges or spikes indicate mechanical irritation, while sustained high-frequency activity correlates with nerve injury and poorer postoperative outcomes. The extent of EMG activity may reflect tumor characteristics (location, size), surgical stage, or history of prior radiation (in case of cranial nerves) [33].

Triggered EMG involves direct electrical stimulation of nerves (or implants such as pedicle screws) to distinguish neural from non-neural structures, with the minimum stimulation threshold used as a predictor of proximity.

In our case, 0.5 mA stimulation was appropriate given the broad surgical field and the large caliber of sciatic fascicles. Unlike cranial nerve surgery, where lower thresholds (0.05–0.1 mA) are typically employed [34], higher values proved both sensitive and specific in our case. Multichannel systems such as the NIM Vital™ (Medtronic) facilitate both sEMG and triggered EMG monitoring. Somatosensory evoked potentials (SSEPs) provide global information on dorsal column–medial lemniscal pathway integrity but are less specific for peripheral-nerve tumors [35].

Ultimately, the choice of resection technique must balance tumor type, intraoperative findings, and functional preservation. Complete tumor excision remains the goal, but this is not always feasible, particularly when the capsule is densely adherent to surrounding muscle [5] or when functional fibers cannot be safely separated [7]. Radical resection in such contexts risks nerve injury, hemorrhage from pelvic venous plexuses, or damage to adjacent viscera [8,9].

In our review (Table A3), 16 of 22 patients (72.7%) underwent complete resection. Two reports did not specify the extent, while in four cases (18.8%), small remnants or capsule fragments were intentionally preserved due to strong adhesions—three in pelvic tumors and one in a rare thigh-extending cluster-type schwannoma. Excluding this outlier, all thigh-level tumors were completely excised.

Postoperative neurological deficits were reported in 5 of 22 patients (22%). Four of these cases involved pelvic tumors, and one distal thigh case recovered fully. This incidence exceeds the 15% reported in the general schwannoma literature [6]. Several factors may explain the discrepancy: (1) giant schwannomas, with mean size ~9.4 cm, are more likely to envelop parent fibers; (2) pelvic tumors are intimately associated with the main trunk, where excision can produce microscopic injury despite gross preservation [6,13]; and (3) anterior extension in the pelvis complicates localization of the intact sciatic trunk, rendering extracapsular removal riskier. Intracapsular resection in such cases requires advanced skill due to proximity to iliac vessels, nerve roots, and pelvic viscera [22]. Functional recovery largely depends on preoperative deficits; sensory symptoms usually resolve quickly, while motor recovery is gradual over months [2]. A study focusing on predictive factors for complications after surgical treatment for schwannoma of peripheral nerves demonstrated the involvement of motor nerves impacting daily functional activities represents an independent risk factor for complications. In his study, motor fiber involvement led to 42% postoperative deficit; however, the majority of patients improved to full recovery or mild disorders with no need for medication [36]. These findings are consistent with our present case, in which the attached fascicle was presumed to belong to the motor component of the nerve. Lakomkin et al. described a case of progressive weakness and paroxysmal pain worsening after partial resection, yet with good postoperative motor recovery [8].

Recurrence occurred in 2 of 22 cases (9%), both associated with malignant transformation. Interestingly, one case recurred despite no visible residual mass, while the other transformed into angiosarcoma but was successfully resected. Thus, true recurrence was observed in only 1 case, underscoring that giant schwannomas rarely recur and size does not predict recurrence risk.

The role of adjuvant therapy in malignant schwannomas remains controversial. Some authors advocate postoperative chemotherapy or radiotherapy to reduce metastatic potential [21,37], while others suggest surveillance following complete resection may suffice [8].

To the best of our knowledge, this is the first reported case of a giant sciatic schwannoma in a patient with Turner syndrome (TS). While previous studies examined estrogen and progesterone involvement in schwannoma and neurofibroma pathogenesis in NF1/NF2, the relationship between TS and schwannomas has not been addressed. Schwann cells synthesize progesterone in response to neuronal signals, which in turn promotes myelination through an autocrine mechanism involving progesterone receptors (PR) [38]. Thus, progesterone availability is essential for Schwann cell function.

Literature shows that PR and ER are upregulated in neurogenic tumors—specifically in neurofibromas. Overdiek et al. demonstrated that neurofibroma-derived Schwann cells express PR and respond to progesterone with increased proliferation [39]. Geller et al. found that 75% of neurofibromas were immunopositive for PR and ER [40]. However, in case of vestibular schwannomas, there are numerous discrepancies related to the role of sexual hormones in schwannoma growth, which are caused predominantly by the different methodologies that have been used by various research groups. In some studies, supporting hormone receptor positivity, the samples were predominantly PR-positive and ER-negative on staining, while others showed no evidence to support the fact that vestibular schwannoma might be a hormone-dependent tumor [41].

In Turner syndrome, gonadal dysgenesis with chronic hormone deficiency (hypoestrogenism, hypoprogesteronism) may lead to diminished receptor expression in neural cells, potentially lowering the overall susceptibility to tumorigenesis. Consequently, the occurrence of a schwannoma in TS should be considered a highly atypical event. This phenomenon may be attributable not only to the disruption of hormonally regulated mechanisms controlling Schwann cell proliferation, supported by some researchers, but also to the involvement of alternative, yet underexplored, proliferative pathways whose significance in schwannoma development may be unmasked precisely by such rare clinical cases that are presented here.

There are studies demonstrating the role of basic fibroblast growth factor (bFGF) and VEGF which interacts with VEGFR receptor tyrosine kinases to drive tumor angiogenesis, fostering a nutrient- and oxygen-rich microenvironment that supports tumor expansion. Elevated VEGF expression has been positively associated with both vestibular schwannoma volume and growth rate [42]. On the other side, schwannoma progression may be decreased by modulating the characteristics of tumor microenvironment by inhibiting cytokines and growth factors (VEGF, bFGF, macrophage colony stimulating factor (M-CSF), IL-34) [41].

On the other hand, in terms of disruption of hormonally regulated mechanisms, it is established that estrogen elicits rapid nonnuclear signal transduction called membrane-initiated steroid signaling (MISS), which induces the mobilization of intracellular calcium, generation of cAMP, modulation of potassium currents, and the stimulation of protein kinase pathways such as PI3K/AKT and ERK [43]. Thus, a decrease in estrogen hormones (TS) may permit other growth factors such as neuregulin-1 to induce proliferative signaling through the ERBB2 and PI3K/MAPK pathway in Schwann cells [41].

Another physiopathological mechanism that could be considered considers hypoestrogenism as a key element in a pro-inflammatory environment by supporting IL-6/TNF-α increase [44]. Furthermore, IL-6 with IL-1 plays a role in NF-κB transcriptional activity, and NF-κB pathway has recently been demonstrated to drive benign tumor progression. Also, merlin, the protein product which is inhibited in NF-2, plays a critical role in regulating Schwann cell proliferation and control nerve development by inhibiting NF-κB activity [45].

Moreover, recent results have indicated that the peripheral nervous system not only synthesizes and metabolizes neuroactive steroids, but it is a target for these molecules. Neuroactive steroids may exert their effects by classical steroid receptors as well as non-classical steroid receptors such as progesterone receptor membrane component 1, GABA, and NMDA. Their levels are different in males and females. Progesterone and its derivates synthesized by the nervous system are able to induce morphological changes, especially in the neuronal growth cones, associated with a rapid reorganization of actin filaments [46].

Another alternative hypothesis is that exogenous steroid hormones may upregulate ER and PR, promoting Schwann cell proliferation and tumor formation, even in the context of baseline deficiency. While plausible, this remains unproven in the literature. A second theory suggests that contraceptives may promote vascular endothelial proliferation, indirectly enhancing tumor vascularity and growth [47].

This article has several limitations inherent to the rarity of the tumor and available literature. Sciatic giant schwannoma is a rare pathology, and the literature presents only isolated case reports and standardized reporting is lacking. The literature has several incomplete or ambiguous reports that were excluded, and there may be a risk of publication bias, especially in terms of follow-up and postoperative evolution. Despite these limitations, the systematic approach applied strengthens the reliability of the findings by ensuring that only cases with clear diagnostic confirmation and adequate clinical information were included.

## 5. Conclusions

In this article, we report a rare giant solitary peripheral nerve schwannoma (15 cm) occurring in a non-NF1, non-NF2 patient. This case is notable in the current literature due to its unique combination of clinical context, medical history, tumor characteristics (size, consistency, location), and intraoperative findings. Despite its extraordinary dimensions and confined location, the tumor produced no clinical symptoms, even at its maximum size. This challenges the prevailing assumption that large tumors invariably lead to severe neurological deficits and highlights instead the critical role of the tumor–nerve interface in determining symptomatology.

Intraoperatively, the tumor initially displaced the nerve stems into a hammock-like configuration, yet over time, it fully enveloped the primary source fiber, rendering dissection impossible. Gross total resection was achieved, and no recurrence was observed at the 6-month follow-up.

The significance of this case is further underscored by the patient’s comorbid Turner syndrome, a condition rarely associated with peripheral nerve sheath tumors. This raises important hypotheses regarding alternative pathophysiological mechanisms beyond the classical sex hormone-dependent models of schwannoma biology. Progesterone and estrogen normally exert trophic effects on Schwann cells, promoting myelination and proliferation via receptor-mediated autocrine pathways. In Turner syndrome, chronic hypogonadism leads to persistent hypoestrogenism and hypoprogesteronism, potentially suppressing these conventional regulatory signals. The occurrence of a giant schwannoma in this hormonal milieu suggests that alternative proliferative mechanisms may compensate. Candidate pathways include upregulation of growth and angiogenic factors such as VEGF and bFGF, which foster tumor vascularization and growth, as well as the activation of inflammatory cascades involving IL-6, IL-1, and NF-κB. The latter is of particular relevance, as NF-κB has been implicated in benign peripheral nerve tumor progression and is normally inhibited by merlin, the NF2 gene product. Similarly, a pro-inflammatory microenvironment with elevated IL-6 and TNF-α may drive NF-κB activity, further supporting schwannoma growth. The peripheral nervous system is also a target for neuroactive steroids, which can modulate Schwann cell morphology and cytoskeletal organization via both classical and non-classical receptors, including progesterone receptor membrane component 1 and GABA receptors. Differences in these pathways between sexes and in the setting of congenital hormone deficiency may account for atypical tumor biology in Turner syndrome. Importantly, this case also illustrates the indispensable role of INM in giant peripheral nerve schwannomas. Our technique allowed reliable identification of functional fascicles. Integration of multimodal INM offers further protection in cases where chronic compression reduces fascicular excitability.

Taken together, this case not only broadens the clinical spectrum of sciatic nerve schwannomas but also highlights how rare comorbidities such as Turner syndrome may unmask non-canonical pathogenic pathways. Future research should focus on integrating molecular studies of angiogenic and inflammatory mediators with clinical observations, to clarify how hormonal and immune imbalances shape Schwann cell tumorigenesis. This perspective underlines the importance of systematic reporting of such rare cases, which—despite their scarcity—offer valuable insights into the complex biology of peripheral nerve sheath tumors. Rather than providing definitive conclusions, this case adds a valuable perspective to the literature and emphasizes the need for further investigation into how sex steroid deficiency and inflammatory dynamics may influence Schwann cell tumorigenesis.

## Figures and Tables

**Figure 1 neurosci-06-00095-f001:**
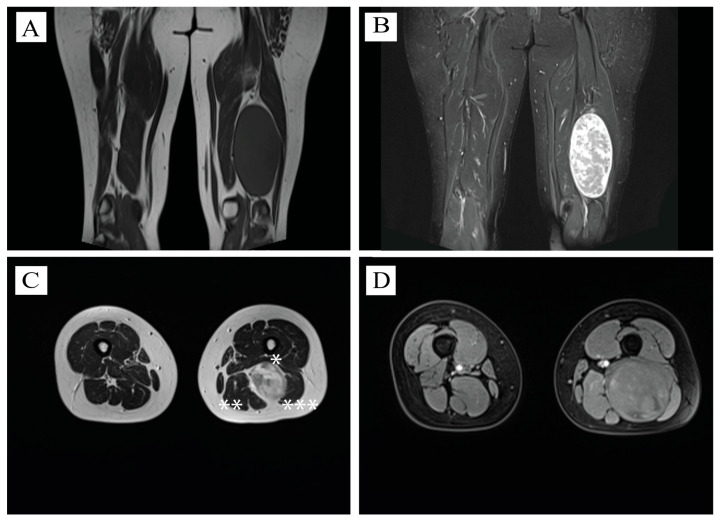
Radiological aspects of the tumor. (**A**) Coronal view of T1WI—sciatic schwannoma localized at the 2/3 inferior part of the left thigh, isointense to adjacent muscle structures, longitudinal diameter—13 cm. (**B**) Coronal view of TIRM signal—suppressing fat signals and enabling clearer visualization of non-fatty tissues, demarcation between normal and pathological tissues. (**C**) Axial view at the level of schwannoma, T2WI—high intensity—axial diameter of 7 cm. *—*Adductor Magnus muscle* compressed by the tumor. **—*Semimembranosus* and *semitendinosus muscle* on the medial site. ***—*Biceps femoris* (*caput longus*)—compressed on the posterior and lateral site. (**D**) Axial T1W1—compressed *Biceps femoris* (*caput longus*) on the upper third of the tumor.

**Figure 2 neurosci-06-00095-f002:**
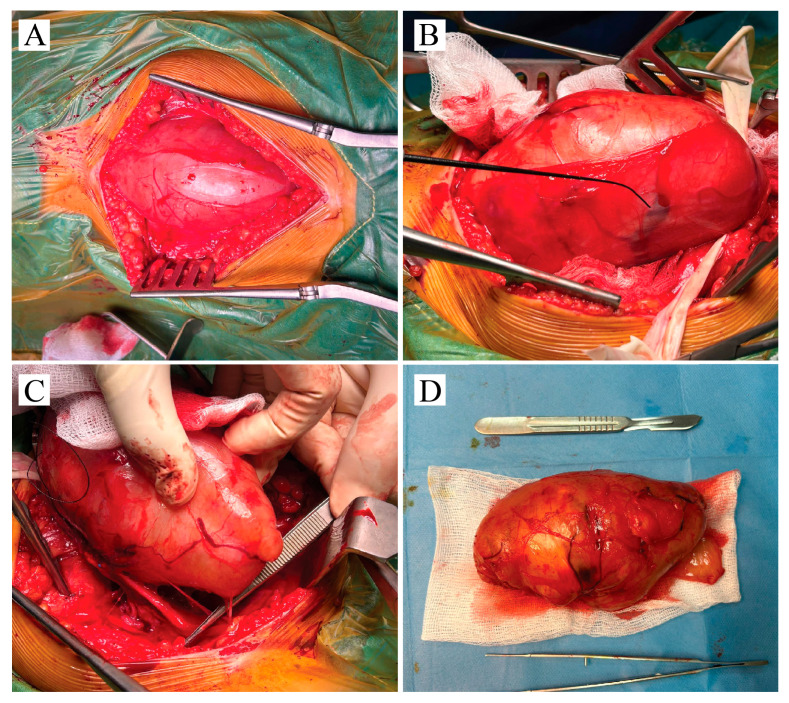
Intraoperative figure with surgical key moments—(**A**) The reddish vascularized tumor capsule is identified directly beneath the subcutaneous fat layer. Surgery continues with stepwise dissection of the tumor from the adjacent fascia. (**B**) The capsule is partially taken off from the tumor. Intraoperative stimulation is used to identify fibers of sciatic nerve within tumor capsule. (**C**) Surgical forceps is used to expose the neural fiber source of the schwannoma. Entire tumor mass enveloped this fiber. (**D**) Tumor mass was resected en-block; dimension are comparative to the scalpel.

**Figure 3 neurosci-06-00095-f003:**
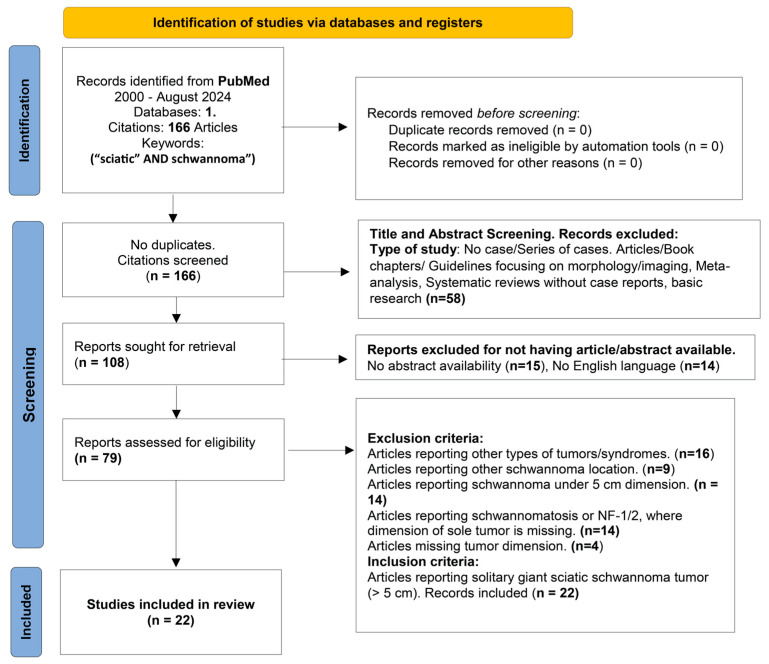
PRISMA 2020 flow diagram for our systematic review following a comprehensive literature search from 2000 to 2024 (October), Articles were retrieved from Pubmed and filtered through screening, retrieval, and application of inclusion and exclusion criteria.

## Data Availability

The raw data supporting the conclusions of this article will be made available by the authors on request.

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
