# Peer review of "Multidisciplinary Management of an Atypical Gigantic Sciatic Nerve Schwannoma: Case Presentation and Systematic Review"

_neurosci, 2025, doi:10.3390/neurosci6040095_

Round 1
Reviewer 1 Report
Comments and Suggestions for Authors
The authors described a multidisciplinary treatment of a giant schwannoma of the sciatic nerve. They also provided a review of the literature.
The case is extremely well described (symptoms, histological examination, surgical intervention)
and illustrated (magnetic resonance imaging and intraoperative images).
The review of the literature provides a comprehensive overview of the pathology.
Author Response
Dear Reviewer,
Thank you for you comments and suggestions.
We appreciate them.
Reviewer 2 Report
Comments and Suggestions for Authors
Nice case report
Add more elements in all the sections of this study
Ameliorate the english language used
Ameliorate the quality of fig 3
Tables 1 and 2 are nice but huge
Please spit them
Add more conclusions
Add mor detailed conclusions
Add more references-add more recent references
Mention the innovation of this study
Comments on the Quality of English LanguageThe English could be improved to more clearly express the research.
Author Response
Dear Reviewer,
We sincerely thank you for the constructive comments that helped us improve the quality and clarity of our manuscript.
- More elements were added to Introduction, Case Report presentation, Materials and Methods, Discussions and Conclusions.
- English was ameliorated.
- Figure 3 Quality was ameliorated
- Tables 1 and 2 were summarized. We have created clearer and more reader-friendly tables.
- We added recent references
- We have highlighted the importance of the study.
We hope this revised form of the manuscript will be considered suitable for publication.
Reviewer 3 Report
Comments and Suggestions for Authors
Lines 80–84: Suggest Including potential pathophysiological connection between gonadal dysgenesis, low estrogen/progesterone, and Schwann cell neoplasia is only superficially addressed for Turner Syndrome. Suggest incorporation of a brief review of molecular drivers (e.g., loss of heterozygosity at neurofibromin loci, NOTCH pathway, hormonal receptor expression in PNSTs). Potential implications for surveillance of such patients.
Lines 121–128; 134: Expand on the indications, limitations, and outcomes of intraoperative neurophysiological monitoring (INM), suggest specify modalities used (e.g., sEMG, triggered EMG, SSEP) with the manufacturer devices (if available), stimulation parameters, and their literature-reported sensitivity for nerve function preservation.
No further comments.
Author Response
Dear Reviewer,
We sincerely thank the reviewer for the thoughtful and constructive comments that helped us improve the quality of our manuscript.
Information suggested were added in the specific paragraphs.
Thank you.
Round 2
Reviewer 2 Report
Comments and Suggestions for Authors
intresting paper
tables a2 and and a3 are intresting but huge please split them
or add some graphics
please amelirate the english language quality
the quality in the indroduction in the discussion in the conlusions is good but you have to add more elements
add also more and more recent references in order to support better the text
Comments on the Quality of English LanguageThe English could be improved to more clearly express the research.
Author Response
Dear Reviewer,
We are pleased to confirm that all recommendations have been carefully addressed as follows:
Both tables have been revised and split/summarize into smaller, more concise units to improve readability.
The entire manuscript has been carefully revised to ameliorate the English language, ensuring fluency and clarity in line with academic standards.
We have further developed Introduction, Discussion, and Conclusions sections by adding more information.
We have incorporated more recent and relevant references, to provide stronger scientific support for the manuscript and place our case in the context of the latest literature.
We trust that these modifications address the reviewer’s comments fully, and we are grateful for the opportunity to improve the manuscript.
We hope that the revised manuscript will now be considered suitable for publication.
Round 3
Reviewer 2 Report
Comments and Suggestions for Authors
Work is real improvment
Compliments to the authors
Please re-design tables A1 and A2
They are Interesting but huge
Author Response
Dear Reviewer,
We are pleased to confirm that your recommendations have been carefully addressed.
Both tables have been revised and summarized into smaller, more concise units to improve readability. Table A1 has been adjusted so that the information is now reduced to checkboxes, making it easier to identify and summarize the data for future analyses. Furthermore, in Table A2 we have summarized and homogenized the imaging features, allowing for a clearer and more consistent analysis. We have also retained the maximum dimension of each lesion as both an inclusion criterion for the review and a key parameter in the discussion section of the study.
We trust that these modifications fully address the reviewer’s comments, and we are grateful for the opportunity to improve the manuscript.
We hope that the revised manuscript will now be considered suitable for publication.